# "Style" Transfer for Musical Audio Using Multiple Time-Frequency Representations

## Abstract

Neural Style Transfer (Gatys et al., 2016) has become a popular technique for generating images of distinct artistic styles using convolutional neural networks. This recent success in image style transfer has raised the question of whether similar methods can be leveraged to alter the "style" of musical audio. In this work, we attempt long time-scale high-quality audio transfer and texture synthesis in the time-domain that captures harmonic, rhythmic, and timbral elements related to musical style, using examples that may have different lengths and musical keys. We demonstrate the ability to use randomly initialized convolutional neural networks to transfer these aspects of musical style from one piece onto another using 3 different representations of audio: the log-magnitude of the Short Time Fourier Transform (STFT), the Mel spectrogram, and the Constant-Q Transform spectrogram. We propose using these representations as a way of generating and modifying perceptually significant characteristics of musical audio content. We demonstrate each representation's shortcomings and advantages over others by carefully designing neural network structures that complement the nature of musical audio. Finally, we show that the most compelling "style" transfer examples make use of an ensemble of these representations to help capture the varying desired characteristics of audio signals.

## 1 Introduction

The problem we seek to explore in this paper is the transfer of artistic "style" from one musical audio example onto another. The definition and perception of an artistic style in visual art images (e.g., impressionist, pointilist, cubist) shown in Figure 1 is perhaps more straightforward than in the case musical audio. For images, a successful style transfer algorithm is capable of generating a novel image whose content information, or what is in the image, is matched as well as its stylistic information, or the artistic approach. In other words, it explores the question, "What would a rendering of scene A by artist B look like?"

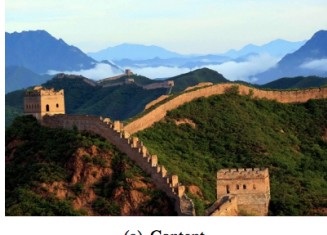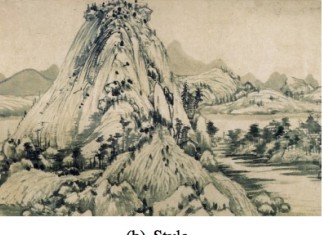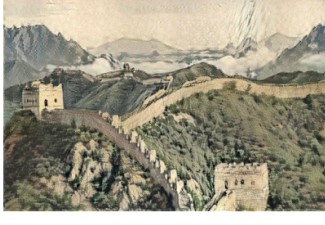

(a) Content        (b) Style        (c) Content + Style

Figure 1: Demonstration of image style transfer courtesy of Jing et al. (2017).

For our work, we similarly set out to develop an algorithm that explores the question, "What would it sound like if a musical piece by ensemble/artist A was performed by ensemble/artist B?" It should be noted that we do not approach the problem according to strict musicological definitions (e.g., melodic, harmonic, rhythmic, and structural elements), as one might proceed if given the musical notation of a composition. We do not presume access to the notation or any music theoretic analysis of a

piece. We are instead interested in transferring the acoustic features related to harmonic, rhythmic, and timbral aspects of one musical piece onto another. Therefore, for the single instance "style" transfer algorithm we propose in this work, it is more accurate to pose the question as "What would a rendering of musical piece A (by artist A) using the harmonic and rhythmic patterns of piece B (by artist B) sound like?" In this paper, we define musical "style" transfer according to this type of audio content transformation, and will henceforth drop the use of quotation marks around "style". In texture generation, we instead ask "What would it sound like for a source musical piece to contain the same musical patterns and higher-order statistics without any of the same local, event-based information?" This can be achieved in the image or audio domain by only optimizing those terms of the loss function of a transfer algorithm associated with style, and not using any loss term associated with content.

Currently, there are two types of approaches to image style transfer. The first method uses a learned generative model to manipulate the representation of the data such that it maintains its original content rendered into a new style. The second class of methods, which we investigate and apply in this paper, are concerned with synthesizing new data that matches the representations of data in a learned model in some specific way. Measuring the accuracy of such algorithms' abilities to transfer style is difficult, since most data is not able to be entirely disentangled into separate content and style components. This is especially true for musical style.

There have been attempts for learning representations of musical style include the use of generative models which use a MIDI representation of audio (Pachet, 2003). The advantages of using this representation are the ability to focus solely on a highly understandable representation of musical information in its harmonic and rhythmic components, but lacks the ability to capture other important sonic information like timbre.

Our approach utilizes many interesting findings from recent research in image style transfer. We suggest that it is possible to use the same style transfer algorithm used for images for musical audio, but best performance requires a careful selection of how content and style is represented, given the task. Figure 2 shows a spectral visualization of how a style transfer result contains both local, event based information from the content piece, while also having the characteristic nature of the style signal, as there is clearly more energy in the higher frequencies. However, it is important to note that despite this visualization in the log-magnitude STFT representation, the audio is ultimately synthesized in the time-domain.

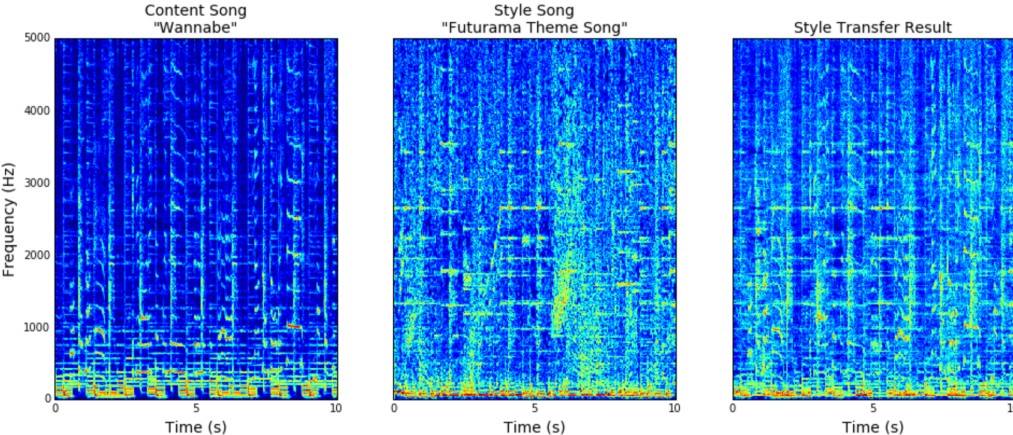

Figure 2: Log Magnitude STFT Representations of an audio style transfer example.

## 2 RELATED WORK

Original attempts at single image style transfer demonstrated successful transfer of artistic style between images through a gradient-based optimization of the convolutional map representations in the Visual Geometry Group (VGG) network (Simonyan & Zisserman, 2014). This work focused on synthesizing preferred images whose layer representations in the VGG network, a convolutional

neural network that rivals human performance on visual object recognition tasks, matching the likeliness of a content image while the statistics of the layer representations of a target style image are also matched. These statistics are typically represented by taking the inner product of the convolutional layer activations in order to compute the gram matrix. The ability to match each of these is expressed through separate content and style loss terms, which can both be used to optimize the pixels of a randomly initialized image (Gatys et al., 2016). The motivation for using the gram matrix to represent style is that local information about the style image is lost, preserving only statistics about the convolutional representation over the entire image. Other work has helped explain the effectiveness of this approach by showing that single image style transfer can be re-written as a Maximum Mean Discrepancy problem with a 2nd-order polynomial distance kernel (Li et al., 2017). It was also shown that the need for a pre-trained network to perform style transfer was not at all necessary. In fact, the ability to recreate the content of the original image was improved when using randomly initialized weights (He et al., 2016). Additionally, Ustyuzhaninov et al. (2016) has shown that very shallow, untrained neural networks with as little as 1 layer are also capable of sufficiently representing artistic style in images.

Ulyanov and Lebedev proposed an algorithm for the transfer of audio style using the log-magnitude STFT of audio with a single random convolutional layer (Ulyanov & Lebedev, 2016). After the novel log-magnitude spectrogram is generated, the Griffin-Lim algorithm (Griffin & Lim, 1984) is used to restore phase information before taking the inverse STFT to recover the time-domain signal. In order to generate these results, only 1 convolutional layer with random, untrained weights was used with a rectified linear activation function. In order to make this model successful, 4096 different convolutional kernels must be used. Also, while spectrograms are typically thought of as F (Frequency bins) x T (Time bins) images with no color channels, this work represented them as 1xT images with F color channels. Therefore, the only convolution taking place is happening in time, while the patterns and relationships between frequencies are localized within the style loss.

More recent advances in generative music models include WaveNet and SampleRNN, which also offer possibilities for style transfer (van den Oord et al., 2016a; Mehri et al., 2016). These auto-regressive models have successfully demonstrated the ability to generate long-term structure in audio signals. SampleRNN uses a multi-hierarchical recurrent neural network trained with Truncated Backpropagation Through Time (TBTT). WaveNet uses causal, dilated convolutions and residual skip-connections, and has the ability to be conditioned on vocal features which gives the ability to generate the same words and speech patterns with different voice characteristics, such as male or female (van den Oord et al., 2016b). There is also a recently proposed WaveNet Autoencoder (Engel et al., 2017), a generative model for raw audio that can be used to manipulate a latent representation of an encoded sound and then regenerate it.

## 3 METHODS

### 3.1 PREVIOUS WORK FROM ULYANOV AND LEBEDEV'S ALGORITHM

In the previous attempt by Ulyanov & Lebedev (2016) at style transfer for audio, a simple convolutional layer with a ReLU non-linearity was used. A single 1-D convolutional layer, whose kernel has a shape that we denote as $1\mathrm{x}K\mathrm{x}F\mathrm{x}N_f$ was used. $K$ represents the length of time bins convolved with the feature, while $F$ and $N_f$ represent the number of frequency bins and filters. These weights are initialized using Glorot Initialization (Glorot & Bengio, 2010). For our notation, we use $x$ to denote the generated log-magnitude STFT, whose feature map, $X$ is the convolved feature map over whichever dimension is being considered for style transfer or texture generation. Following this, we use $s$ and $c$ to denote the log-magnitude STFT representations for the style and content audio, respectively, whose feature maps are represented by $S$ and $C$, respectively. The $L_2$ distance between the target audio and the content audio's feature maps summed over all spatial indices $i$ and channels $j$ is used for content loss expressed in Equation 1.

$$L_{Content}(C, X) = \frac{1}{2} \sum_{i,j} (C_{ij} - X_{ij})^2 \qquad (1)$$

In representing style, it is ideal that local information about where musical events happen is lost, but information about how they happen in relation to each other is maintained. To represent the style,

the inner-product, or gram matrix of the convolutional feature map is used. The inner-product of the vectorized feature maps $X$ and $S$, denoted by $W$ and $G$ respectively, are calculated using Equations 2 and 3 for a filter $i$ and another filter $j$ is used to represent style.

$$G_{i,j} = \sum_k S_{ij} S_{jk} \qquad (2)$$

$$W_{i,j} = \sum_k X_{ij} X_{jk} \qquad (3)$$

The style loss, $L_{Style}$, calculated as the sum of the $L_2$ distances between $G$ and $W$ over all pairs of filters $i$ and $j$ in $N_f$, is given in Equation 4.

$$L_{Style}(G, W) = \frac{1}{N_f^2} \sum_{i,j} (G_{ij} - W_{ij})^2 \qquad (4)$$

The total loss is represented by Equation 5, which uses parameters $\alpha$ and $\beta$ to measure the importance of transferring each type of style.

$$L(X, C, S) = \alpha L_{Content} + \beta L_{Style} \qquad (5)$$

All network weights are unaltered during optimization of the $L$. Only the log magnitude STFT representation of the target audio is adjusted.

## 3.2 Proposed Audio Representations

We introduce an extended version of the algorithm proposed in Ulyanov & Lebedev (2016) with different versions of two additional log-magnitude spectrogram representations described below in order to address the shortcomings of this algorithm as it applies to musical audio. In particular, Ulyanov & Lebedev (2016) characterizes and represents the timbral style of musical audio, which we can defined in this work as the short time envelope and harmonic statistics of the audio signal, but fails to capture information which is either rhythmic or harmonic. We propose using the Mel Spectrogram to better capture rhythmic information, and using the Constant Q Transform (CQT) Spectrogram in a 2-D convolutional neural network to represent harmonic style. While we use the Mel spectrogram to help represent rhythmic information, the true benefit of this representation is that the compressed channel axis allows for a deeper dilated network structure with a much longer receptive field in time.

### 3.2.1 Mel Spectrogram

While this simple, single layer design is effective in representing short-term harmonic and timbral features of audio, its most obvious fault is its inability to effectively represent longer-term features of the audio signals, which we refer to as the rhythmic components. In order to increase the size of the kernel that spans the temporal dimension of a spectrogram, we need to also decrease the size of the filter dimension in, or the number of frequency channels. Without this, the size of our computation graph becomes very large, and the time needed for synthesis is greatly increased. We argue that rhythmic information is still preserved when compressing this frequency dimension. We propose using the same algorithm on a Mel scaled spectrogram of the log-magnitude STFT, which we refer to as the Mel spectrogram version of the signal, $x_{Mel}$. The Mel scale provides a mapping from the perceived Mel center frequency and the actual measured frequency, $f$, as shown in Equation 6 below. The mapping can be used to create a filter bank for projecting the magnitude STFT onto a perceptually optimal smaller number of channels.

$$M(f) = 1125 \ln(1 + \frac{f}{700}) \qquad (6)$$

Because the Mel spectrogram decreases spectral resolution of the STFT in a perceptually uniform manner, it has been a popular choice for state of the art neural networks trained on large corpuses of

musical audio (Choi et al., 2016; Wyse, 2017). However, instead of using 2-D convolutions on the Mel spectrogram like this work, we propose treating the mel-frequency axis as a channel axis rather than a spatial axis like is done for the STFT representations.

While a large number of filters in a convolutional kernel are still needed to represent style from this representation, the significantly reduced number of frequency bins means the number of our parameters in the kernel can be much smaller. In order to get a much larger receptive field of audio, we use a much longer kernel and a multi-tiered dilated non-causal convolutional structure modeled after the WaveNet auto-encoders (Engel et al., 2017). This architecture makes use of dilated convolutions with residual, skip-connections in its encoding stage to vastly increase the receptive field. In this way, our model actually can be thought of as a generalization of this encoding model structure, but we have inserted the Mel spectrogram transformation in front of the initial layer which normally receives raw audio. While the original convolutional kernel used in Ulyanov & Lebedev (2016)'s implementation only had a receptive field of about 30 ms, this structure is capable of representing up to 4 seconds of audio with only 2 residual blocks for audio sampled at 22.05 kHz and $N_{DFT} = 2048$. Additionally, since the dimensionality of the representation in the channel dimension is decreased by at least 2, we find that we can use a much smaller number for $N_f$, which reduces the size of the computation graph needed for the same receptive field and decreases computation time. Following the WaveNet auto-encoder architecture, we only use dilated convolutions if more than one residual block is used, starting with a dilation rate of 2, and doubling again for each additional residual block. We compute the style loss from each layer in this neural network structure, and the content loss only using the last layer. In practice, we use between 16 (128:1 reduction) and 512 Mel filters (4:1 reduction) that are triangularly shaped and span from 0 Hz to 22.05 kHz.

### 3.2.2 CONSTANT Q TRANSFORM (CQT) SPECTROGRAM

While the STFT and Mel spectrogram allow us to represent both short and long term musical statistics well, neither representation is capable of representing musically relevant frequency patterns, such as chords and intervals, along the frequency axis that can be shifted linearly. In order to achieve this, we need to use a 2-D convolution over a spectral representation. We have chosen the CQT spectrogram (Schörkhuber & Klapuri, 2010) since it is capable of representing transposition in musical pitch as a simple shift along the (warped) frequency axis. In this way, an activation for a particular harmony at a note, n, should be close in value to the same harmonic structure being present at any other note within +/-6 semitones. Similarly to the Mel spectrogram, it has often been chosen as a representation for neural networks for this reason. While in some ways the CQT spectrogram is worse than the Mel spectrogram in terms of perceptual reconstruction quality due to the frequency scaling in lower frequencies, it is the most natural representation for 2-D convolutional kernels to use to represent joint time-frequency spatial features.

We implement the "pseudo-CQT", which is obtained by projecting the magnitude STFT signal onto a wavelet filter bank with Constant Q frequency scaling, rather than the recursive sub-sampling method described in Schörkhuber & Klapuri (2010), which has the benefit of modeling the transformation as a fully differentiable operation on the raw time-domain audio. In order to achieve key invariance for musical content, we use a total of 86 bins with 12 bins per octave spanning from the notes C1 to D8, and use max-pooling along the convolved frequency axis if key-invariance is desired. This allows features from the target audio to match the features of the content and style audio from multiple keys. As the width of the pooling increases, the representation becomes more key-invariant, and the distortion of the content representation increases. This filter bank is similarly logarithmic to the Mel filter bank in the higher frequency range (> 1kHz), but it oversamples the lower frequencies in comparison.

### 3.3 PARALLEL ARCHITECTURE AND TIME DOMAIN SYNTHESIS

We generalize the formulation described in section 3.1 for the content and style loss to yield 6 total possible loss terms (content and style for all 3 networks) to be used during optimization: $L_{Content,STFT}$ ; $L_{Content,Mel}$; $L_{Content,CQT}$; $L_{Style,STFT}$; $L_{Style,Mel}$; $L_{Style,CQT}$. For each content loss, use the $L_2$ loss of the final layers' representations. For the style loss terms, we use the sum of all $L_2$ loss of the gram matrix representations of each layer in the network as described in Section 3.1. For the CQT network representations, the inner-product of both the temporal and frequency axes is used since both frequency and time invariance is desired. In practice, we make

use of all possible terms, using up to 5 at once. Since there are up to 5 different objectives in our proposed solution, it can be difficult to discover the optimal weighting for a style transfer example. In order to alleviate this process, we propose initializing each loss term's coefficient such that the magnitude of the gradient is 1 for all loss terms as shown in Equation 7, inspired by Li et al. (2017). We then control the scaling of the then normalized loss terms using the parameter, $\Gamma_{Type,Net}$, as shown in Equation 8.

$$\beta'_{Type,Net} = \frac{1}{\left\| \frac{\partial L_{Type,Net}}{\partial x_{time}} \right\|}, Type \in \{Content, Style\}, Net \in \{STFT, Mel, CQT\} \quad (7)$$

$$\beta_{Type,Net} = \Gamma_{Type,Net}\beta'_{Type,Net} \quad (8)$$

The objective function to be minimized is the sum of all of these losses scaled by $\beta_{Type,Net}$ as expressed in Equation 9.

$$L(X, C, S) = \sum_{\substack{Type \in \{Content, Style\} \\ Net \in \{STFT, Mel, CQT\}}} \beta_{Type,Net}L_{Type,Net} \quad (9)$$

The previous method computes the log-magnitude STFT as a pre-processing step and, after optimizing the result, used the Griffin-Lim algorithm (Griffin & Lim, 1984) to reconstruct phase. We, instead, propose a method for estimating phase simultaneously with the optimization of the STFT representation. This transformation is fully differentiable to the raw time domain audio signal, meaning if overlapping time windows are used, phase information can be approximated. We use gradient descent based optimization algorithm Limited Memory BFGS (L-BFGS-B) for its superior performance in non-linear optimization problems including image style transfer. Recent work has shown that BFGS is not only capable of reconstructing phase in time-domain signals from magnitude STFT representations, but it converges faster and can reconstruct phase on some signals on which Griffin-Lim fails (Decorsière et al., 2015). Because of this, we choose to model the transformation from the time-domain target input to the log-magnitude STFT as well as the Mel and CQT representations extracted as a projection of the target audio onto different bases as symbolic combinations of differentiable convolutional and dense layers in our computation graph. This allows us to optimize the time-domain audio from each spectral representation network's gradient in one stage without suffering major phase distortion. Another advantage of optimizing the target audio directly in the time domain is that it allows also for neural representations of the time-domain audio to be optimized simultaneously. We see this as being outside of the scope of the work being presented here, however.

While each of these representations has complementary advantages and disadvantages, we can still use all 3 representations in parallel neural networks to capitalize upon the advantages from each. This is entirely possible through the ensemble representation of each network shown in Figure 3 below.

### 3.4 PROPOSED KEY-INVARIANT CONTENT REPRESENTATION

Prior work in style transfer proposed that higher layer neural representations of images captured the content more than style, where content is any information relevant to the classification task (Gatys et al., 2016). We seek to create a similarly meaningful content representation, which maintains general information about musical events, without specifics like key and timbre. Key-invariance is desired since modern popular music is often composed without an emphasis on the specific key, and instead the harmony is commonly defined as its relation to the "one", which in Western music can be any of 12 musical keys. For these reasons, we argue that key-invariance is a crucial feature for a representation of musical content. We propose and test 3 methods for examining invariance to musical key between the content and style audio sources.

For the first, we suggest using a Mel spectrogram content representation with only a few channels. This has the effect of taking a wide-band average of the audio signal, capturing the envelope of the audio signal. This type of representation captures only the rhythmic hits and the overall phonemes of words being sung. While prominent rhythmic and timbral shape information is preserved, all information about key is lost since the signal is being averaged over large regions of frequency. In

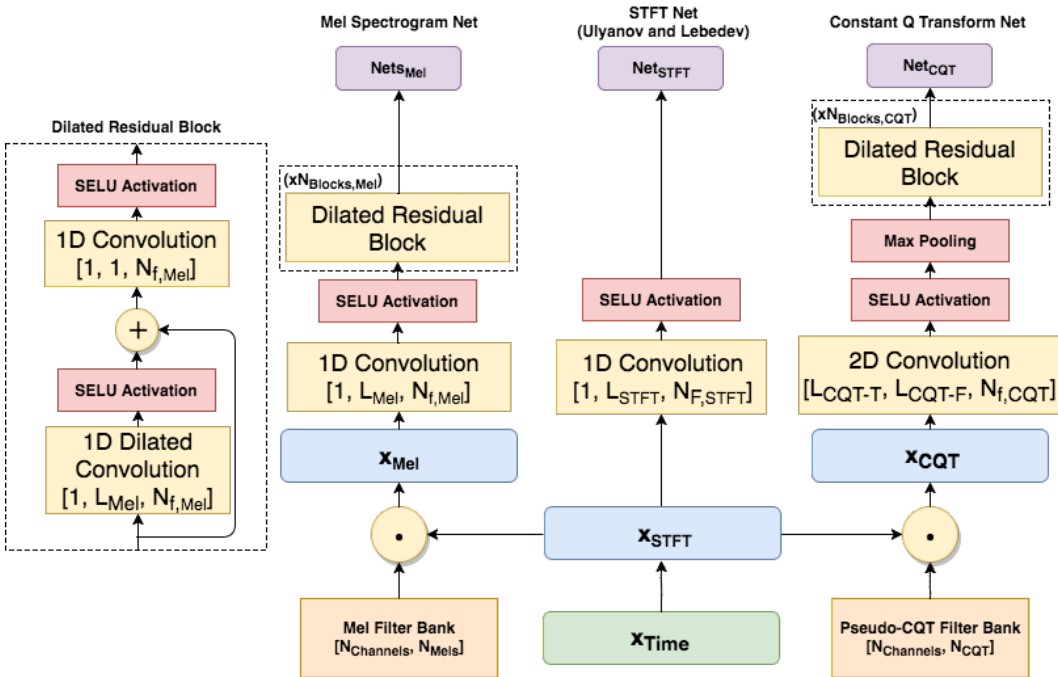

Figure 3: Full diagram of proposed architecture used for musical style transfer.

practice, we use anywhere between 8 and 16 Mel bins with a max Mel bin frequency of 11.025 kHz. Reducing further obscures phonetic information and any distinctive timbral components and adding additional Mel bins leaks information about the key of the song.

For the second key-invariant content representation, we wish to use the convolved 2-D feature map of the CQT spectrogram. We choose a convolutional kernel that is 11x11, meaning it spans just under one octave at each stride. We use a max-pooling layer in order to make this content representation key-invariant, and we use max-pooling along the convolved frequency axis of the convolutional feature map. This representation ensures that content that has been shifted in key, or shifted along the warped frequency axis of the CQT spectrogram, will yield a similar representation as the non-shifted version of the same signal. However, it is important to note that this max-pooling layer is only beneficial for making $L_{Content,CQT}$ key-invariant, since the $L_{Style,CQT}$ uses the inner product of the spatial dimension of the data, making it already key-invariant. In comparison with the key-invariant Mel spectrogram representation, this representation better preserves melodic and harmonic information without maintaining local key information. In practice we use a stride of 2 for the pooling, however, a greater stride could be used for increased invariance at the cost of greater information loss. It is important to note that pooling occurs on the convolutional layer versions of the representation, not the raw CQT itself. Pooling in this fashion would destroy harmonic information, while pooling on extracted patterns spanning enough frequency allows for these same patterns (i.e. a major third interval) to be placed at any fundamental frequency within the width of semi-tones defined by the pooling size.

Finally, for the best approximation, we recommend combining both of these loss terms to encapsulate the information from each key-invariant representation.

## 3.5 SELU ACTIVATION FUNCTION FOR BETTER PERFORMANCE

Previous work in image style transfer has shown that using instance normalization layers can improve quality of the synthesized images (Ulyanov et al., 2016). Unfortunately, when we attempted to implement these results for audio, these layers also seemed to introduce a noticeable amount of noise to the target audio. Instead, we use newly proposed self-normalizing non-linear activation functions, SeLUs (Klambauer et al., 2017), in place of ReLUs (Nair & Hinton, 2010) since they have both

desirable non-linear properties of activation functions as well as the property of allowing network activations to have 0 mean and unit variance with LeCun Normal initialization (LeCun et al., 1998). We find that using this activation function increases the quality of the synthesized audio, reduces convergence time, and makes finding an optimal weighting for the style and content terms more consistent from example to example. Since we don't use any type of dense neural network layers in our proposed networks, we demonstrate that it is possible to use different lengths of content and style audio with the same convolutional kernels and still obtain a valid style loss. To achieve this, we simply divide by the size of the convolutional feature map being used prior to computing the difference in the style loss calculation.

## 4 EXPERIMENTS AND RESULTS

We propose 3 experiments for showing the ability for our algorithm to improve the transfer of musical style. First, we show how using the Mel spectrogram representation of style is able to better capture the musical statistics of the audio through texture generation. We do so by examining what happens when we generate musical textures, meaning only style loss with no content. We also examine how the combined representation of the Mel and CQT representations of content offer a frequency invariant representation. Finally, we examine the quality of our algorithm for a variety of musical style transfer examples using both quantitative and qualitative assessment.

For hyper-parameters, we find that using 4096 filters for the STFT network, 1025 for the Mel network, and 256 for the CQT network in each convolutional layer to be sufficient. We use kernel sizes of 1x11 (~25 ms), 1x50 (~1 second at first layer), and 11x11 (11 semitones x ~25 ms) for the STFT, Mel, and CQT networks respectively as well. However, for residual layers in the Mel network, we use a 1x25 kernel. Since we use SELUs, we also use LeCun Normal weight initialization (LeCun et al., 1998).

All experiments were conducted using a Tensorflow implementation of the ensemble model described in the previous section. We have released code used for the experiments.[1] Additionally, we also have a collection of musical style transfer examples, including all of the examples used for the results shown below.[2]

### 4.1 MUSICAL TEXTURE GENERATION

While textures synthesized using prior methods exhibit a lack of rhythmic structure, we show the use of residual dilated non-causal convolutions on the Mel spectrogram works well for achieving a large receptive field and better capturing rhythmic statistics. In order to measure the rhythmic style similarity of a texture to its source audio, we compute the Kullback-Leibler (KL)-divergence of the Inter-Onset Interval Length Distributions. we detect onsets using Librosa's onset detector, which picks peaks based on the differences of energy in adjacent time frames of a spectrogram (McFee et al., 2015). While increasing the receptive field to a much longer length improves this measure of rhythmic similarity, it is important to show that this is not introducing the effect of copying sections of the original source audio. In order to verify that this is not occurring, we also show that the maximum cross-correlation value between the time-domain audio waveforms are not significantly affected by the length of this field. We denote $max(\Phi_{XY})$ as the maximum cross-correlation value between two signals, $X$ and $Y$. This helps justify that the source and texture are still significantly different, so that the texture isn't simply a distorted duplicate of the source. Additionally, we show that the mean local autocorrelation has more consistent hierarchical rhythmic structure at different lag times. Figure 4 summarizes these results.

### 4.2 TESTING KEY INVARIANCE

In order to test the quality of key-invariance we suggest that it must be possible to use songs in two different keys for style and content, and be able to reconstruct the same content in the key of the style audio. We synthesized 13 different transpositions from a MIDI score of 4 second clips from "The Star Spangled Banner" and performed style transfer experiments where the original key version is used as

---

[1] https://github.com/anonymousiclr2018/Style-Transfer-for-Musical-Audio
[2] https://anonymousiclr2018.github.io/jekyll/update/2017/05/20/musical-style-transfer.html

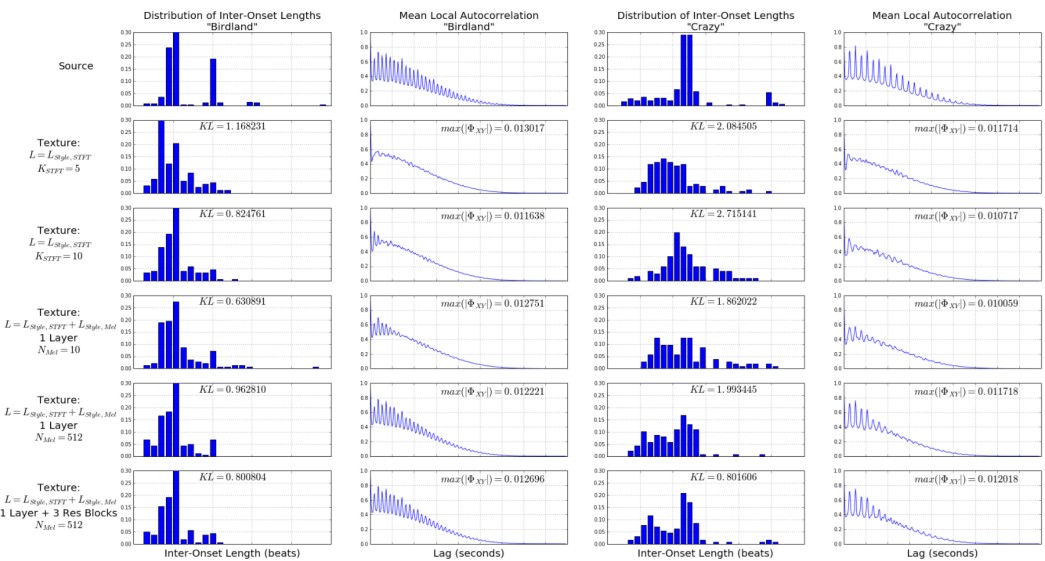

Figure 4: Columns 1 and 3: comparison of inter-onset lengths distribution and KL divergence from the source distribution for a texture generation example as the effective receptive field increases in time. Columns 2 and 4: mean local autocorrelation plots showing the increase in hierarchical rhythmic structure of the audio without any significant increase in the maximum cross-correlation value, $max(\Phi_{XY})$.

content, and the versions which are transposed +/-6 semitones are used for style. We used both the mean squared error of the log-magnitude STFT representation of the result and the style used as a metric for content key-invariance. We choose this metric because the log-magnitude STFT contains most information about original time-domain audio signal, and we argue that being able to reconstruct the exact style signal should be possible if the content and style are the same, but in different keys. Figure 5 shows the full results for all experiments. Our results confirm that changing key between style content has less of an effect on our proposed key-invariant content representations. The results confirm that for all cases where the content and style pieces are in different keys, all of the proposed key-invariant representations have lower error than the normal STFT representations. While there is no clear best representation of these three proposed versions according to this quantitative metric, we observe that using both the $CQT$ and $Mel$ representations captures different kinds of key-invariant information and combining these representations yields the best sounding results. Samples from this experiment are included in the supplement.

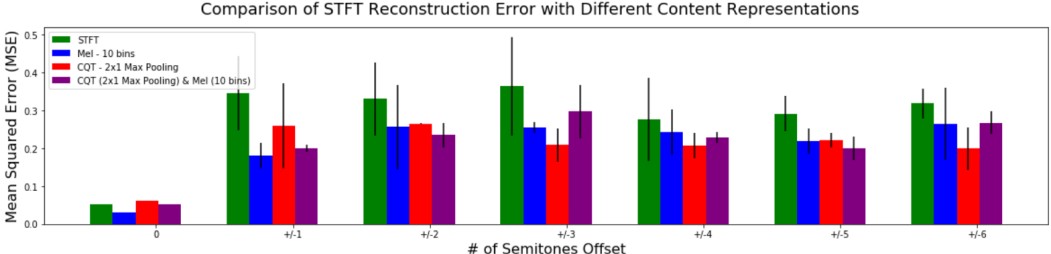

Figure 5: Comparison of the error with different content-based representations for a task where the content and style audio is exactly the same except for key. The x-axis represents varying semi-tone offsets in musical representation. The first point on the left of 0 semi-tone offset represents the identity style transfer task, where both content and style are exactly the same signals. We plot the error in the log-magnitude STFT representations to show the overall signal error.

### 4.3 QUALITATIVE COMPARISON OF POSSIBLE LOSS TERM CONFIGURATIONS

We tested the full style transfer algorithm with different combinations of loss terms and hyper-parameters for a diverse set of examples. We found that there is not one set of loss terms that works best for all content-style transfer pairs, as the best set of loss terms is usually dependent on the nature of the specific style transfer task. However, we are able to demonstrate specific cases where using only the STFT loss terms fails, and significant improvement is achieved with the introduction of the other objectives based on perceptual audio representations. All examples use content and style with tempos within 10 beats per minute (bpm) of each other. We summarize our findings for a variety of loss content and style representation formulations in Table 1.

Table 1: Summary of the effects of using different possible loss term formulations. We find that using both of our proposed content and style loss formulations improves the example quality from the simple STFT architecture proposed in Ulyanov & Lebedev (2016) for many examples. The table references specific song pairs for content and style in musical transfer examples found at the URL given previously.[2]

| Content Representations | Style Representations | | |
|---|---|---|---|
| | STFT | STFT + Mel (1 Layer, Many Channels) | STFT + Mel (1 Layer + 2 Residual Layers, Many Channels) + CQT (1 Layer + 3 Residual Layers, Many Channels) |
| STFT | (1) Fastest, works well for examples where content and style are highly similar (See "Imperial March"/"Star Spangled Banner") | (2) Works well on most examples where key is aligned (See "Ghostbusters"/"Get Lucky" and "In Da Club"/"Africa") | (3) Needed when very long-term musical information is considered (See "Bohemian Rhapsody"/"Scary Monsters and Nice Sprites") |
| Mel (1 Layer, Few Channels) & CQT (1 Layer, 2x1 Max Pooling) | (4) Works best for singing style transfer (e.g. to a capella) (See "God Only Knows"/"Isn't She Lovely") | (5) Works well on almost all examples, improves some examples from 2 (See "Love-shack"/"Psychosocial") | (6) Works best for most difficult examples, where 1-5 fail (See "S.O.S."/"I Love It") |

We notice that using Mel spectrogram representations for achieving a larger receptive field in time greatly helps transfer style in certain cases where the long-term nature of the style audio is complex, and important to recognizing the nature of the musical style. Using both Mel and CQT representations for content greatly helps for cases where the nature of the content and style audio are very different from each other, even in cases where the content and style are in the same key. The abstracted content representation is better able to re-implement content with the musical attributes of the style audio without simply laying 2 different sounding examples over each other. This increase in performance is best displayed in examples of singing style transfer. Using a higher number of Mel channels and residual blocks will give the behavior similar to a "mash-up" of 2 audio examples, where portions of the style audio are placed over the content audio. Simply reducing the number of Mel channels eliminates this behavior so that the $L_{Style,STFT}$ has high resolution frequency information over a short time span, representing the timbral and short-time harmonic features only, and $L_{Style,Mel}$ uses a much longer time span with low frequency resolution, representing long-term musical attributes like rhythmic structure. When we use the Mel spectrogram content representation, we notice that the wide-frequency band nature of the synthesized audio sometimes has an undesirable, "breathy" quality to it. This problem is most evident when there are vocals. In order to restrain this effect, we add the

$L_1$ penalty to the log-magnitude STFT to enforce the synthesized audio to have a sparse spectrum. We find that this helps eliminate the evenly distributed nature of the content part of the synthesized audio.

## 5 CONCLUSIONS AND FUTURE WORK

We introduce several improvements for performing musical style transfer on raw audio through the utilization of multiple audio representations. Our contributions can be summarized as follows: First, we have demonstrated that using additional representations of Mel and CQT spectrograms with accompanying neural structure improve in many cases the capture of musically meaningful style information. Secondly, we have proposed a novel, key-invariant content representation for musical audio. Finally we have shown that despite using log-magnitude spectrograms to capture the content and style information, we are still able to synthesize a target audio waveform in the time domain using the backpropogation of the STFT.

While our proposed content representations work for audio in different keys, there still is no representation for tempo invariance. Other future work may include using learned generative models to perform musical style transfer and trying to perform style transfer entirely in the time-domain. This or the use of complex weights may be able to help improve representation of phase information in neural representations.

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
