# OpenReview forum: "“Style” Transfer for Musical Audio Using Multiple Time-Frequency Representations"
_ICLR.cc/2018/Conference — Reject_

### Official Review · AnonReviewer2 · 2017-11-22
**Not enough substantial science, results not compelling**

**Rating:** 4
**Confidence:** 4

**Review:**

This paper studies style transfer for musical audio, and largely proposes some additions to the framework proposed by Ulyanov and Lebedev.  The changes are designed to improve the long-term temporal structure and harmonic matching of the stylized audio.  They carry out a few experiments to demonstrate how their proposed approach improves upon the baseline model.

Overall, I don't think this paper provides sufficiently novel or justified contributions compared to the baseline approach of Ulyanov and Lebedev.  It largely studies what happens when a different spectrogram representation is used on the input, and when somewhat different network architectures are used.  These changes are interesting, but don't provide a lot of additional information which I believe would be interesting to the ICLR community.  They seem better suited for an (audio) signal processing venue, or a more informal venue.  In addition, the results are not terribly compelling.  If the proposed changes (which are heuristically, not theoretically motivated) resulted in huge improvements to sound quality, I might be convinced.  More concretely, the results are still very far away from being able to be used in a commercial application (in contrast with image style transfer, whose impressive results were immediately commercially applied).  One reason I think the results remain bad is that the audio signal is still fundamentally represented as a phase-invariant representation.  Even if you backpropagate through the time-frequency transformation, the transformation itself discards phase, and so various signals (with different phase characteristics) will appear the same after transformation.  I believe this contributes the most to the fact that the resulting audio sounds very artifact-ridden and unrealistic.  If the paper had been able to overcome this limitation, I might be more convinced, but as-is I don't think it warrants acceptance at ICLR.

Specific comments:

- The description of Ulyanov & Lebedev's algorithm in 3.1 is confusingly structured.  For example, the sentence "The L2
distance between the generated audio and the content audio’s feature maps..." is basically a concatenation of roughly 6 thoughts which should be separated into different sentences.  The location of the equations (1), (2) do not correspond to where they are introduced in the text.  In addition, I don't understand how S and C are generated.  It is written that S and C are the "log-magnitude feature maps for style and content".  But the "feature maps" X are themselves a log-magnitude time frequency representation (x) convolved with the filterbank.  So how are S and C "log-magnitude feature maps"?  Surely you aren't computing the log of the output of the filterbank?  More equations would be helpful here.  Finally, it would be helpful to provide an equation both for G and W instead of just saying that W is analogously defined.
- I don't see any reason to believe that a mel-scaled spectrogram would better capture longer time scales or rhythmic information.  Mel-scaling your spectrogram just changes the frequency axis to a mel scale, which makes it somewhat closer to our perception; it does not modify the way time is represented in any way.  In fact, in musical machine learning tasks usually swapping between CQT and mel-scaled spectrograms (with a comparable number of frequency bins) has little effect, so I don't see any compelling reason to use one for "rhythm".  You need to provide strong empirical or theoretical evidence to back up your claim that this is a principled approach.  Instead, I would expect that your change of convolutional structure (to the dilated convolutions, etc) for the "mel spectrogram" branch of your network would account more heavily for stronger modeling of longer timescales.
- You refer to "WaveNet auto-encoders" and cite van den Oord et al.  The original wavenet paper did not propose an auto-encoder structure; Engel et al. did.
- "neither representation is capable of representing spatial patterns along the frequency axis" What do you mean by this?  Mel or linear-frequency (or CQT) spectrograms exhibit very strong patterns along their frequency axis.
- The method for automatically setting the scales of the different loss terms seems interesting, but I can't find anywhere a description of how you apply each of the beta terms.  Are they analogous to the alpha and beta parameters in equation (4)?  If so, it appears that gamma is shared across each beta term; this would mean that changing the value of gamma simply changed the scale of all loss terms at once, which would have no effect on optimization.
- "This is entirely possible though the ensemble representation" typo, through -> through
- That instance normalization causes noisy audio is an interesting empirical result, but I'm interested in a principled explanation of why this would happen.
- "the property of having 0 mean and unit variance" - you use this to describe the SeLU nonlinearity.  That's not a property of the nonlinearity, it's a property of the activations of different layers when using the nonlinearity (given correct initialization).
- How are the "Inter-Onset Interval Length Distributions" computed?  How are you getting the onsets, etc?
- " the maximum cross-correlation value between the time-domain audio waveforms are not significantly affected by the length of this field" - there are many ways effective copying could happen without the time-domain cross-correlation being large.

---

### Official Review · AnonReviewer3 · 2017-11-27
**Shows promise, but should focus on cover song creation; needs more detail on method**

**Rating:** 6
**Confidence:** 4

**Review:**

This paper describes improvements to a system described in a blog post for musical style transfer.  Such a system is difficult to evaluate, but examples are presented where the style of one song is applied to the content of another.  These audio examples show that the system produces somewhat reasonable mixtures of two songs, but suggest that if the system instead followed the (mostly) accepted rules for cover song generation, it could make the output much more pleasant to listen to.  Additional evaluation includes measuring correlations between style songs and the output to ensure that it is not being used directly as well as some sort of measure of key invariance that is difficult to interpret.  The paper does not completely define the mathematical formulation of the system, making it difficult to understand what is really going on.

The current paper changes the timbre, rhythm, and harmony of the target content song.  Changing the harmony is problematic as it can end up clashing with the generated melody or just change the listener's perception of which song it is.  I suggest instead attempting to generate a cover version of the content song in the style of the style song. Cover songs are re-performances of an existing (popular) song by another artist.  For example, Jimi Hendrix covered Bob Dylan's "All along the watchtower" and the Hendrix version became more popular than the original.  This is essentially artist A performing a song by artist B, which is very similar to the goal of the current paper.  Cover songs almost always maintain the lyrics, melody, and harmony of the original, while changing the timbre, vocal style, tempo, and rhythmic information.  This seems like a good way to structure the problem of musical style transfer.  Many systems exist for identifying cover songs, see the relevant publications at the International Society for Music Information Retrieval (ISMIR) Conference.  Few systems do something with cover songs after they have been identified, but they could be used for training a system like the one proposed here, if it could be trained.

Another musically questionable operation is pooling across frequency in the constant-Q transform representation.  In western music, adjacent notes are very different from one another and are usually not played in the same key, for example C and C#.  Thus, pooling them together to say that one of them is present seems to lose useful information.  As part of the pooling discussion, the paper includes an investigation of the key-invariance of the model.  Results from this are shown in figure 5, but it is difficult to interpret this figure.  What units is the mean squared error measured in?  What would be a big value?  What would be a small value?  What aspects of figure 5 specifically "confirm that changing key between style [and] content has less of an effect on our proposed key-invariant content representations"?

Section 3.1, which describes the specifics of the model, is confusing.  What exactly are S, C, W, and G?  What are their two dimensions indexed by i and j?  How do you compute them from the input?  Which parameters in this model are learned and which are just calculated?  Is there any training or is L(X,C,S) just optimized at test time?

Finally, the evaluation of the texture generation part of the algorithm could be compared to existing texture generation algorithms (there are several) such as McDermott & Simoncelli (2011, NEURON), which even has code available online.



Minor comments
--------------

p2: "in the this work" typo

p2: "an highly" typo

p2: "The first method... The latter class of methods" confusing wording.  Is the second one a different method or referring back to the previous method?  If it's different, say "The second method..."

p7: Please describe kernel sizes in real units (e.g., ms, Hz, cents) as well as numbers of bins



After revision/response
--------------------------------
The revisions of the paper have made it clearer as to what is going on, although the description of the algorithm itself could still be described more mathematically to really make it clear.  It is more clear what's going on in figure 5, although it could also be further clarified whether the green bars are showing the distance between log magnitude STFTs of the transposed "style" snippets and the untransposed "content" snippets directly and so provide an upper bound on the distances. My overall rating of the paper has not changed.

---

### Official Review · AnonReviewer1 · 2017-11-27
**Some points need clarification, but overall good.**

**Rating:** 7
**Confidence:** 3

**Review:**

Summary
-------
This paper describes a method for style transfer in musical audio recordings.
The proposed method uses three spectral representations to encode rhythm (Mel spectra), harmony (pseudo-constant-Q), and content (STFT), and the shared representation designed to allow synthesis of the time domain signal directly without resorting to phase retrieval methods.
Some quantitative evaluation is presented for texture synthesis and key invariance, but the main results seem to be qualitative analysis and examples included as supplemental material.


Quality
-------

I enjoyed reading this paper, and appreciate the authors' attention to the specifics of the audio domain.
The model design choices make sense in general, though they could be better motivated in places (see below).
I liked the idea of the rhythm evaluation, but again, I have some questions about the specific implementation.
The supplementary audio examples are somewhat hit or miss in my opinion, and a bit more qualitative analysis or a listener preference study would strengthen the paper considerably.


Clarity
-------

While for the most part, the writing is clear and does a good job of describing the representations used, there are a few parts that could be made more explicit:

- Section 3.2: The motivation for using the Mel spectrum to capture rhythm seems shaky.  Each frame has the same temporal resolution as the input STFT representation, so any ability to
  capture rhythmic content should be due to down-stream temporal modeling (dilated residual block, in the model).  This does not necessitate or suggest a Mel spectrum, though dimensionality
  reduction is probably beneficial.  It would be good to provide a bit more motivation for the choices made here.

- Section 3.2.1: the description of the Mel spectrogram leaves out a few important details, such as the min/max frequencies, shape of the filters, and number of filters.

- Section 3.2.2: the "pseudo-CQT" described here is just a different log-frequency projection of the STFT (and not constant-Q), and depending on the number of parameters, could be quite
  redundant with the Mel spectrum.  A bit more detail here about the parametrization (filter shapes, etc) and distinction from the Mel basis would be helpful.

- Section 3.3: I didn't completely follow the setup of the objective function.  Is there a difference gamma parameter for each component of the model?

- Section 3.4: What is the stride of the pooling operator in the harmony component of the model?  It seems like this could have a substantial impact on any key-invariance properties.

- Section 4.1: the idea to measure IOI distributions is clever, but how exactly is it implemented?  Does it depend on a pre-trained onset detector?  If so, which one?  I could imagine
  texture synthesis preserving some aspects of rhythm but destroying onset detection accuracy due to phase reconstruction (whooshiness) problems, and that does not seem to be controlled in
  this experiment.

- Figure 4: how is Phi_{XY} defined?  The text does not convey this clearly.

- Section 4.2: why is MSE of the log-stft a good metric for this problem?  Changing the key of a piece would substantially change its (linear) spectrum, but leave it relatively constant in
  other representations (eg Mel/MFCC, depending on the bin resolution).  Maybe I don't entirely understand what this experiment is measuring.

- General note: please provide proper artist attribution for the songs used in the examples (eg Figure 2).


Originality
-----------

The components of the model are not individually novel, but their combination and application are compelling.
The approaches to evaluation, while still somewhat unclear, are interesting and original to the best of my knowledge, and could be useful for other practitioners in need of ways to evaluate
style transfer in music.



Significance
------------

This paper will definitely be of interest to researchers working on music, audio, or creative applications, particularly as a proof of concept illustrating non-trivial style transfer outside
of visual domains.

---

### Author Response · Authors · 2018-01-05
**Thank you for your feedback!**

We have updated our paper according to your comments. We have tried to clarify the design of our algorithm.

One thing we tried to emphasize is that the use of the Mel spectrogram is not in its natural ability to capture any type of rhythmic information, but more for its ability to capture information that is still rhythmically relevant with a greatly reduced number of channels. Since a 1D convolutional kernel tensor has a signification number of parameters (# Time bins x # filter in x # filters out), we wanted to create a network that could have more span in time at the cost of less filters in. This lets us represent much more time with a smaller computation graph. As reviewer 2 pointed out, the dilated convolutional network structure we propose is more responsible for the increase in receptive field of time.

Another point we'd like to clarify is that pooling for the CQT representations happens in the layer representations, not directly on the CQT spectrogram. We agree that pooling directly on the CQT would destroy harmonic information, but propose instead that pooling extracted frequency patterns that could be shifted along this axis would promote key invariance in representations.

Thanks again for your time.

---

### Decision · Program_Chairs · 2018-01-29
**ICLR 2018 Conference Acceptance Decision**

**Decision:**

Reject

**Comment:**

The paper extends an existing work with three different frequency representations of audios and necessary network structure modifications for music style transfer.
It is an interesting study but does not provide "sufficiently novel or justified contributions compared to the baseline approach of Ulyanov and Lebedev". Also the revisions can not fully address reviewer 2's concerns.